# Bridging the research gap in conflict-affected countries: A multi-institutional study on medical students' research involvement and barriers in Yemen

Mohamed Baklola[1]*☯, Naji Al-bawah[2]*☯, Mohammed Al-Shehari[3],
Amira Yasmine Benmelouka[4], Amjed Al-Jahmi[2]*, Ahmed A. Khailah[2], Hend O. Al-Sabri[2],
Khaled Badr[5], Saif Alaribi[6], Jafar A. Mohammed[7], Hadeel Alashwal[8], Mohammed Badi[9],
Ehab Sharyan[2]

**1** Faculty of Medicine, Mansoura University, Mansoura, Egypt, **2** Faculty of Medicine, Sana`a University, Sana'a, Yemen, **3** General Surgery Department, Faculty of Medicine and Health Sciences, Sana'a University, Sana'a, Republic of Yemen, **4** Faculty of Medicine, University of Algiers, Algiers, Algeria, **5** Faculty of Medicine, University of Science and Technology, Sana`a, Yemen, **6** Faculty of Medicine, Emirates International University, Sana`a, Yemen, **7** Faculty of Medicine and Human Science, University of Aden, Aden, Yemen, **8** Faculty of Medicine, Alyemenia University, Sana`a, Yemen, **9** Faculty of Medicine-EDC, University of Khartoum, Khartoum, Sudan

☯ These authors contributed equally to this work.
* Mohamedbaklola2000@gmail.com (MB); Najialbawah@gmail.com (NA); amjedaljahmi157@gmail.com (AA-J)

## Abstract

### Background

Undergraduate research engagement plays a pivotal role in promoting evidence-based medical practice, particularly in low-resource and conflict-affected settings such as Yemen. However, multiple structural and academic barriers often limit students' ability to actively participate in research. This study aimed to assess medical students' research knowledge, attitudes, participation, and perceived barriers in a multi-institutional setting in Yemen.

### Methods

A cross-sectional survey was conducted among undergraduate medical students from ten universities in Yemen using a structured online questionnaire assessing research knowledge, attitudes, participation, and perceived barriers. Data were collected between April and July 2025. Statistical analyses included descriptive statistics, independent t-tests, one-way analysis of variance, and linear regression to identify factors associated with research knowledge and attitudes.

**Data availability statement:** The datasets generated and analyzed during this study are publicly available in the Figshare repository at https://doi.org/10.6084/m9.figshare.30927329.

**Funding:** The author(s) received no specific funding for this work.

**Competing interests:** The authors have declared that no competing interests exist.

## Results

A total of 1,387 students participated in the study, with a mean age of 23 years; 70.7% were male. Although most students (79.8%) agreed that research should be an integral part of medical education, only 20.5% reported having published a research paper, and 41.5% had never attempted to write one. The mean research knowledge score was low (2.8 out of 10), with no significant difference by gender. Fifth-year students demonstrated the highest knowledge scores (mean = 3.2, SD = 2.0; $p = 0.003$). Academic performance was a significant predictor of both research knowledge and attitudes ($p < 0.001$). The most frequently reported barriers were insufficient training in research methods (81.0%), inadequate research facilities and resources (76.0%), limited research opportunities (74.2%), and insufficient faculty guidance (72.3%).

## Conclusion

Participating medical students demonstrated generally positive attitudes toward research; however, their research knowledge and active involvement remained limited. Persistent institutional and structural barriers, including limited research training, mentorship, and infrastructure, may hinder meaningful research engagement. Strengthening research training and institutional support within undergraduate medical education may help improve research engagement in resource-limited and conflict-affected settings.

## Background

Medical research serves as a cornerstone of modern healthcare systems by informing clinical practice, guiding health policies, and driving innovation [1]. However, global research output remains strikingly uneven, with the vast majority originating from high-income countries. In contrast, low-resource and conflict-affected nations, such as Yemen, contribute relatively little to the global scientific landscape despite bearing a disproportionate share of disease burden and health system challenges [2]. This disparity may contribute to underrepresentation of low-resource settings in evidence-based decision-making [3]. One important contributor to this disparity is the limited integration of research training within undergraduate medical education in many low-resource settings. Strengthening research engagement during undergraduate medical training is therefore increasingly recognized as an important strategy for improving long-term research capacity in such contexts.

One of the most sustainable and cost-effective ways to address this gap is to promote research engagement among undergraduate medical students [4]. It also fosters a research-oriented mindset that can influence future career paths and increase the likelihood of pursuing clinician-scientist roles [5]. However, in many conflict-affected and resource-constrained countries, this potential remains largely untapped due to systemic barriers such as lack of institutional support, inadequate mentorship,

limited access to training in research methodology, and insufficient infrastructure [6]. Understanding how these barriers affect students' research engagement is essential for informing educational strategies and strengthening research capacity within medical training programs.

Studies from the Arab region suggest that while students generally exhibit favorable attitudes toward research, their actual knowledge and participation remain low. Evidence from neighboring countries in the region indicates that limited structured research training and insufficient mentorship opportunities are common challenges affecting undergraduate research engagement [7]. In Yemen specifically, only a few small-scale, institution-specific studies have investigated students' research experiences [8,9]. These studies have consistently highlighted positive attitudes but also revealed a lack of practical engagement, poor knowledge, and numerous barriers such as heavy academic workloads, lack of time, and minimal faculty involvement [8]. However, most existing studies have been limited to single institutions, making it difficult to obtain a comprehensive national picture of students' research engagement.

Given Yemen's ongoing conflict, humanitarian crisis, and fragile health and education systems, a comprehensive understanding of medical students' research capacity is urgently needed. In such settings, strengthening undergraduate research engagement may contribute to developing a workforce capable of generating locally relevant evidence to inform healthcare policies and clinical practice. This study addresses this gap by examining research knowledge, attitudes, participation, and perceived barriers among undergraduate medical students from ten universities in Yemen. By providing the largest multi-institutional assessment of medical students' research engagement in the country to date, this study offers valuable evidence to inform educators, policymakers, and academic institutions seeking to strengthen undergraduate research capacity and address persistent research inequities in conflict-affected and resource-limited settings.

## Methods

### Study design and location

This study employed a cross-sectional study design and was conducted among undergraduate medical students from ten universities in Yemen. The participating institutions included both public and private universities located in different regions of the country. The study aimed to assess medical students' research knowledge, attitudes, participation, and perceived barriers to research engagement in a multi-institutional academic context. Data were collected through an online survey administered between April and July 2025.

### Sample size

The sample size for this study was determined based on the primary outcome of interest, which was the mean research knowledge score among undergraduate medical students. A previous study conducted in Jordan assessing knowledge, attitudes, practices, and perceived barriers toward medical research reported a mean knowledge score of 3.76 with a standard deviation of 1.78 [7]. Because no prior large-scale studies assessing research knowledge among Yemeni medical students were available, estimates from this study were used to inform the sample size calculation. Using this estimate, the minimum required sample size was calculated using standard sample size calculation methods for continuous outcomes, assuming a 95% confidence level and 90% statistical power. This resulted in a minimum required sample size of 370 participants.

### Sampling and data collection approach

A non-probability convenience sampling approach was used to recruit participants from ten medical universities in Yemen. The survey was administered electronically using Google Forms. The survey link was disseminated through commonly used student communication platforms, including Telegram and WhatsApp groups managed by student representatives and university-affiliated channels. Members of the research team also facilitated distribution within their respective institutions to ensure participation from students across different academic years.

Eligibility criteria included current enrollment in a Yemeni undergraduate medical program. Participation was voluntary, and informed consent was obtained electronically prior to survey initiation. Responses were collected anonymously, and no personally identifiable information was recorded. Students were able to complete the survey at their convenience during the data collection period.

## Study tools

The study utilized a structured, English-language questionnaire adapted from a previously validated instrument developed in a similar context in Jordan to assess medical students' knowledge, attitudes, practices, and perceived barriers related to research engagement [7]. To ensure clarity and contextual appropriateness, the questionnaire was reviewed by senior faculty members and piloted among a small group of medical students prior to full dissemination; no major modifications were required.

The questionnaire consisted of five sections. The first section gathered demographic information, including age, gender, academic year, GPA, and the intention to pursue postgraduate studies abroad. The second section explored students' prior research involvement, such as participation in research projects or presentations during secondary school and medical school, using five Yes/No questions. The third section measured attitudes toward research using six statements rated on a 5-point Likert scale ranging from "strongly disagree" to "strongly agree," with two negatively worded items reverse-coded to maintain scoring consistency. The fourth section assessed perceived barriers to conducting research, including access to training, research infrastructure, and faculty mentorship, using eight Likert-type items. The final section assessed research knowledge using ten multiple-choice questions, each with four response options.

The full set of knowledge assessment items is provided in Supplementary material 1 to allow evaluation of content validity. The questionnaire demonstrated acceptable internal consistency, with Cronbach's alpha values of 0.78 for the attitude scale and 0.81 for the perceived barriers scale, indicating satisfactory reliability.

## Ethical approval and consent to participate

This study adhered to the ethical principles outlined in the Declaration of Helsinki and its subsequent revisions. Ethical clearance was granted by the Institutional Review Board (IRB) of the Faculty of Medicine and Health Sciences at Sana'a University (Approval No. 2396). Participation was entirely voluntary, with all data collected anonymously to maintain participant confidentiality. The study's objectives and procedures were clearly explained on the introductory page of the online survey. Prior to commencing the questionnaire, all participants provided electronic informed consent. They were also assured of their right to refuse or withdraw from the study at any time without facing any penalties or negative consequences.

## Statistical analysis

All statistical analyses were conducted using IBM SPSS Statistics version 27. Descriptive statistics were used to summarize demographic characteristics and key study variables. Continuous variables, including age, research knowledge scores, and attitude scores, were presented as means and standard deviations (SD), while categorical variables such as gender and academic year were summarized as frequencies and percentages. The normality of continuous variables was assessed using the Shapiro–Wilk test and visual inspection of histograms. As the distributions of knowledge and attitude scores approximated normality, parametric statistical tests were applied. Independent samples t-tests were used to compare mean scores between two groups (e.g., gender), while one-way analysis of variance (ANOVA) was used to compare mean scores across multiple groups (e.g., academic year, GPA categories, and intention to pursue postgraduate studies abroad). Where significant differences were identified through ANOVA, post-hoc analyses were conducted to determine specific group differences. Multiple linear regression analyses were performed to identify factors associated with research knowledge and attitude scores. Variables including age, academic year, and academic performance were entered into the

regression models based on theoretical relevance and evidence from previous literature. Statistical significance was set at a p-value < 0.05 for all analyses.

## Results

### Demographic and academic characteristics

The study included 1,387 participants, with a mean age of 23 years (SD = 2.4) (Table 1). The majority of participants were male (70.7%). Participants were distributed across academic years, with the highest representation in the 4th year (29.7%). In terms of academic performance, 36.8% of students reported having an Excellent academic score, followed by 34.2% with a Very Good score. Most students attended public high schools (55.6%), and a significant majority (62.7%) expressed an intention to travel abroad after medical school. Regarding research experience, 41.5% of students had never attempted to write a research paper, while 20.5% had published their research in journals. Among students who reported attempting to write a research paper, 23.6% participated during medical school only, while 18.0% reported participation during both school and medical school.

### Attitudes toward research

Medical students demonstrated generally positive attitudes toward research (Table 2). A strong majority (79.8%) agreed or strongly agreed that research should be part of the medical curriculum. Similarly, 73.2% agreed or strongly agreed that

**Table 1. Demographic and academic characteristics of the study participants.**

| Variable | | n (%), N = 1,387 |
|---|---|---|
| **Age,** mean (SD) | | 23.0 (2.4) |
| **Sex** | Female | 407 (29.3%) |
| | Male | 980 (70.7%) |
| **Academic year** | 1st | 96 (6.9%) |
| | 2nd | 234 (16.9%) |
| | 3rd | 302 (21.8%) |
| | 4th | 412 (29.7%) |
| | 5th | 272 (19.6%) |
| | 6th | 71 (5.1%) |
| **Academic score** | Excellent | 510 (36.8%) |
| | Very Good | 474 (34.2%) |
| | Good | 346 (24.9%) |
| | Pass | 57 (4.1%) |
| **High school program** | Private | 616 (44.4%) |
| | Public | 771 (55.6%) |
| **Intention to travel abroad after medical school** | Not Sure | 322 (23.2%) |
| | No | 196 (14.1%) |
| | Yes | 869 (62.7%) |
| **Have you ever attempted to participate in writing a research paper?** | No | 575 (41.5%) |
| | Yes, in school | 235 (16.9%) |
| | Yes, in university | 327 (23.6%) |
| | Yes, in school and university | 250 (18%) |
| **Have you ever published your research paper in journals?** | No | 1103 (79.5%) |
| | Yes | 284 (20.5%) |

**Table 2. Percentages of medical students' answers on the attitude scale.**

| Attitude items | Strongly disagree | Disagree | Neutral | Agree | Strongly agree |
|---|---|---|---|---|---|
| 1. Research should be part of Bachelor of Medicine and Bachelor of Surgery (MBBS) curriculum | 17 (1.2%) | 63 (4.5%) | 200 (14.4%) | 540 (38.9%) | 567 (40.9%) |
| 2. Research will not help in better understanding of subject | 252 (18.2%) | 493 (35.5%) | 287 (20.7%) | 261 (18.8%) | 94 (6.8%) |
| 3. It is an extra burden to do research | 67 (4.8%) | 230 (16.6%) | 446 (32.2%) | 448 (32.3%) | 196 (14.1%) |
| 4. Research will help one's clinical practice later | 23 (1.7%) | 95 (6.8%) | 254 (18.3%) | 592 (42.7%) | 423 (30.5%) |
| 5. It is a waste of time and it disturbs studies | 239 (17.2%) | 415 (29.9%) | 300 (21.6%) | 300 (21.6%) | 133 (9.6%) |
| 6. Your research record should be an important criterion for acceptance in residency | 39 (2.8%) | 160 (11.5%) | 371 (26.7%) | 549 (39.6%) | 268 (19.3%) |

**Notes:** Data are presented as n and row %.

research would help their clinical practice later. However, a notable proportion (46.4%) agreed or strongly agreed that research is an extra burden, and 31.2% viewed it as a waste of time that disrupt studies. Interestingly, 58.9% of students agreed or strongly agreed that research records should be an important criterion for residency acceptance.

## Knowledge and attitude scores by demographic and academic characteristics

The mean total knowledge score was 2.7 (SD = 1.6) for females and 2.8 (SD = 1.7) for males, with no significant difference between genders (p = 0.20) (Table 3). Similarly, total attitude scores were comparable between females and males (24.4 ± 2.8 vs. 24.6 ± 2.9, p = 0.20).

Significant differences in knowledge and attitude scores were observed across academic years (p = 0.003 and p = 0.02, respectively). Fifth-year students demonstrated the highest mean knowledge score (3.2 ± 2.0), whereas first-year students reported the highest mean attitude score (25.4 ± 3.4). Academic performance was significantly associated with attitude scores (p < 0.001), with students reporting excellent academic performance demonstrating the highest mean attitude score (24.8 ± 2.7), although differences in knowledge scores across academic performance categories were not statistically significant (p = 0.18).

No significant differences in knowledge or attitude scores were observed based on students' intention to travel abroad after medical school or prior attempts to participate in research writing (p > 0.05). However, students who reported having published a research paper demonstrated significantly higher knowledge and attitude scores compared with those who had not published (knowledge: p = 0.03; attitude: p = 0.01).

## Predictors of knowledge and attitude scores

Regression analysis revealed that age, academic year, and academic score were significant predictors of total knowledge scores (p < 0.05) (Table 4). Specifically, higher academic years (Beta = 0.1, p < 0.001) and better academic performance (Beta = 0.08, p < 0.001) were associated with increased knowledge scores. For total attitude scores, academic score was the only significant predictor (Beta = 0.09, p < 0.001).

## Barriers to research

Students identified several barriers to research participation, with the most prominent being insufficient training in medical research, which was agreed or strongly agreed upon by 81.0% of respondents (Table 5). Additionally, 74.2% of students highlighted the lack of sufficient research opportunities as a significant obstacle, while 72.3% pointed to the lack of stimulation and support from faculty members. Another major barrier was the lack of adequate research facilities and resources, acknowledged by 76.0% of participants. Time constraints also played a critical role, with 59.6% of students agreeing or

**Table 3. Comparison of total knowledge and total attitude scores by demographic and academic characteristics.**

| Variable | Knowledge score | | Attitude score | |
|---|---|---|---|---|
| | Mean (SD) | p-value | Mean (SD) | p-value |
| **Gender** | | | | |
| Female | 2.7 (1.6) | 0.2 | 24.4 (2.8) | 0.2 |
| Male | 2.8 (1.7) | | 24.6 (2.9) | |
| **Academic year** | | | | |
| 1st | 2.4 (1.4) | **0.003*** | 25.4 (3.4) | **0.02*** |
| 2nd | 2.4 (1.4) | | 24.2 (2.6) | |
| 3rd | 2.8 (1.7) | | 24.6 (3.0) | |
| 4th | 2.7 (1.5) | | 24.5 (2.9) | |
| 5th | 3.2 (2.0) | | 24.9 (2.7) | |
| 6th | 2.6 (1.7) | | 23.8 (2.7) | |
| **Academic score** | | | | |
| Excellent | 2.9 (1.8) | 0.18 | 24.8 (2.7) | **0.001*** |
| Very Good | 2.7 (1.7) | | 24.6 (3.0) | |
| Good | 2.5 (1.4) | | 24.2 (2.9) | |
| Pass | 2.7 (1.7) | | 23.9 (3.1) | |
| **Intention to travel abroad after medical school** | | | | |
| Not Sure | 2.8 (1.7) | 0.16 | 24.3 (3.0) | 0.13 |
| No | 2.4 (1.4) | | 24.2 (2.7) | |
| Yes | 2.8 (1.7) | | 24.8 (2.8) | |
| **Have you ever attempted to participate in writing a research paper, either in school or in university?** | | | | |
| No | 2.7 (1.7) | 0.8 | 24.6 (3.0) | 0.14 |
| Yes, in school. | 2.6 (1.4) | | 24.6 (2.7) | |
| Yes, in university. | 2.9 (1.9) | | 24.5 (3.0) | |
| Yes, in school and university. | 2.6 (1.5) | | 24.5 (2.6) | |
| **Have you ever published your research paper in journals?** | | | | |
| No | 2.7 (1.6) | **0.03*** | 24.5 (2.9) | **0.01*** |
| Yes | 3.0 (1.9) | | 24.7 (2.8) | |

**Note:** * indicates statistically significant values <0.05. Independent t-test used for two-group comparisons; one-way ANOVA used for comparisons across more than two groups.

**Table 4. Linear regression models for predictors of knowledge and attitude scores.**

| Outcome | Predictor | Beta | p-value | Model $R^2$ |
|---|---|---|---|---|
| Knowledge score | Age | 0.06 | 0.01* | 0.02 |
| | Academic year | 0.10 | < 0.001* | |
| | Academic performance | 0.08 | < 0.001* | |
| Attitude score | Academic performance | 0.09 | < 0.001* | 0.01 |

**Note:** * indicates statistically significant values < 0.05. Although predictors were statistically significant, the low $R^2$ values indicate that the models explain only a small proportion of variance.

**Table 5. Students' opinions on barriers to research.**

| Barrier items | Strongly disagree | Disagree | Neutral | Agree | Strongly agree |
|---|---|---|---|---|---|
| 1. Insufficient training in medical research | 13 (0.9%) | 66 (4.8%) | 185 (13.3%) | 566 (40.8%) | 557 (40.2%) |
| 2. Lack of sufficient research opportunities | 16 (1.2%) | 86 (6.2%) | 256 (18.5%) | 641 (46.2%) | 388 (28.0%) |
| 3. Lack of stimulation and support from faculty members | 26 (1.9%) | 84 (6.1%) | 274 (19.8%) | 492 (35.5%) | 511 (36.8%) |
| 4. Lack of adequate research facilities and resources | 22 (1.6%) | 84 (6.1%) | 227 (16.4%) | 523 (37.7%) | 531 (38.3%) |
| 5. Lack of sufficient funding | 20 (1.4%) | 81 (5.8%) | 261 (18.8%) | 529 (38.1%) | 496 (35.8%) |
| 6. Insufficient time for medical students | 43 (3.1%) | 161 (11.6%) | 357 (25.7%) | 409 (29.5%) | 417 (30.1%) |
| 7. Faculty members do not have sufficient time | 65 (4.7%) | 216 (15.6%) | 360 (26.0%) | 430 (31.0%) | 316 (22.8%) |
| 8. Insufficient number of faculty members | 79 (5.7%) | 171 (12.3%) | 331 (23.9%) | 467 (33.7%) | 339 (24.4%) |

**Notes:** Data are presented as n and row %. Likert scale ranged from strongly disagree (1) to strongly agree (5).

strongly agreeing that they do not have sufficient time for research, and 53.8% noting that faculty members also lack sufficient time to support student research activities.

## Discussion

This multi-institutional study assessed the knowledge, attitudes, practices, and perceived barriers related to medical research among undergraduate medical students in Yemen. Overall, the findings indicate that although many students reported favorable attitudes toward research, their research knowledge and active participation were limited. While students exhibit generally favorable attitudes toward research, these attitudes were accompanied by relatively low knowledge scores and limited publication experience. Such attitudes must be complemented by foundational knowledge and applicable skills to translate into effective research engagement [10,11].

In this study, the mean knowledge scores were low, echoing findings from previous research in similar settings [7,12–14]. For example, a study at Hadhramout University showed that although a vast majority of students had positive views toward research (90.9%), the majority lacked adequate knowledge, reflected by a mean score of just 2.72 out of 10 [15]. These findings suggest that positive perceptions alone may not be sufficient to promote effective research engagement without structured training and institutional support.

One underlying reason may be the absence of institutional incentives that encourage student research. In contrast to some high-income and Western countries, medical education systems in Yemen do not formally require research output for residency or postgraduate admission. In contrast to some high-income and Western countries, medical education systems in Yemen do not formally require research output for residency or postgraduate admission, which may reduce students' motivation to prioritize research activities [7]. Additionally, unlike reports from other Arab countries, our findings did not demonstrate gender-based differences in knowledge or attitudes, where females have sometimes shown greater engagement and academic drive [7,16]. This may reflect the overarching influence of systemic and structural barriers that affect students regardless of gender.

Notably, students in more advanced academic years, particularly those in their fifth year, and those with higher GPAs performed better in terms of research knowledge and attitudes. This trend mirrors data from other countries in the region [10,16–18] and suggests that greater academic exposure and clinical training may increase familiarity with research concepts and perceived relevance. However, although academic year and performance were statistically significant predictors, the low explanatory power of the regression models indicates that these factors account for only a small proportion of the variability in research knowledge and attitudes, suggesting that additional institutional or environmental factors not captured in the present study may also influence research knowledge and attitudes. Students planning to pursue

postgraduate training abroad may also perceive research experience as a competitive advantage, potentially increasing their interest in research participation [7]. These findings may provide useful evidence for educators and policymakers seeking to strengthen research training and institutional support for undergraduate medical students in Yemen.

Despite their apparent interest, students identified several structural and institutional hindrances to research participation. The most frequently reported barriers included insufficient training in research methodologies, limited mentorship from experienced faculty, inadequate research opportunities, and poor access to research facilities and resources. In the Yemeni context, these barriers are closely intertwined with the ongoing conflict, which has disrupted higher education systems through faculty migration, interrupted salaries, damaged infrastructure, limited institutional funding, and restricted access to international academic journals and databases. Similar barriers have been reported in studies from Saudi Arabia and other regional settings, emphasizing the central role of mentorship and institutional capacity in shaping undergraduate research engagement [13]. Broader issues, such as poor access to data, underdeveloped infrastructure, financial limitations, and even restricted internet access, further exacerbate the problem in Yemen [11,19].

Psychological readiness, including traits like adaptability and resilience, also appears to influence research engagement. The concept of cognitive flexibility has been linked to greater tolerance of research challenges and fewer negative perceptions of barriers [20]. Recent evidence suggests that students with lower cognitive flexibility tend to view research more negatively and perceive greater obstacles [21]. In resource-limited and unstable environments, fostering psychological adaptability may represent a potential area for further research when exploring factors influencing student research engagement [22].

Furthermore, when students do engage in research, their involvement tends to be limited to basic tasks such as data entry or collection, with limited exposure to critical elements like study design, statistical analysis, or academic writing [15]. Majumdar and colleagues highlighted this imbalance and advocated for early incorporation of structured methodology training to enhance competency [23]. Integrating formal research training earlier within the medical curriculum may help address these gaps and promote more meaningful student involvement [24].

Research involvement during undergraduate years has been positively correlated with future academic productivity and a sustained interest in scholarly activities. Students with early research experience often demonstrate higher levels of knowledge, more constructive attitudes, and a greater likelihood of pursuing research-oriented careers [25,26]. Consequently, the introduction of structured research electives, accessible mentorship programs, and protected research time within medical curricula could substantially enhance both the quality and quantity of undergraduate research output [27,28].

Models from Europe and North America offer valuable guidance, where elective research modules have successfully improved students' confidence and sustained their academic interests [11,29]. Successful initiatives typically incorporate dedicated research time, accessible mentors, clearly defined objectives, and alignment with students' academic and career goals. Adapting such models to conflict-affected and resource-limited settings, while accounting for local constraints, may contribute to strengthening future research capacity among medical graduates capable of contributing to context-specific health research and evidence-based practice [30].

## Limitations

This study has several important limitations that should be considered when interpreting the findings. First, the use of a non-probability convenience sampling approach may have introduced selection bias, as students with a preexisting interest in research or greater access to online platforms may have been more likely to participate. As a result, the findings may not be fully generalizable to all undergraduate medical students in Yemen. Second, data were collected using a self-administered online questionnaire, which may be subject to recall bias or social desirability bias, particularly for self-reported measures of research experience and attitudes. Third, the cross-sectional study design limits the ability to infer causal relationships or assess changes in research engagement over time. Finally, although the questionnaire was adapted from a previously validated instrument, formal validation within the Yemeni context was limited, and cultural or

institutional differences may have influenced the interpretation of certain items. Future research employing longitudinal designs, probability-based sampling strategies, and mixed-methods approaches is warranted to further validate and expand upon these findings.

## Conclusion

This multi-institutional study provides insight into the research engagement of undergraduate medical students in Yemen, a country facing significant challenges related to ongoing conflict and resource limitations. While students generally demonstrated positive attitudes toward research, their overall research knowledge and active participation remained limited, with several structural and institutional barriers identified, including inadequate research training, limited mentorship, and insufficient research infrastructure. These findings suggest the potential value of integrating structured research training, mentorship opportunities, and institutional support mechanisms within undergraduate medical curricula. However, the findings should be interpreted cautiously due to the study's methodological limitations, including the cross-sectional design and convenience sampling approach. Future efforts aimed at strengthening undergraduate research capacity may help support the development of locally relevant health research in conflict-affected and resource-limited settings.

## Supporting information

**S1 File. Supplementary material 1.** Knowledge assessment questionnaire. This file contains the full set of knowledge assessment items used in the study to evaluate content validity, along with the attitude and perceived barriers scales included in the survey instrument.
(DOCX)

## Author contributions

**Conceptualization:** Mohamed Baklola, Naji Al-bawah, Ehab sharyan, Amira Yasmine Benmelouka, Saif Alaribi.

**Data curation:** Mohamed Baklola, Naji Al-bawah, Amira Yasmine Benmelouka, Ahmed A. Khailah, Saif Alaribi, Jafar A. Mohammed, Hadeel Alashwal.

**Formal analysis:** Mohamed Baklola, Naji Al-bawah, Ehab sharyan, Ahmed A. Khailah, Khaled Badr, Saif Alaribi, Jafar A. Mohammed, Hadeel Alashwal.

**Funding acquisition:** Naji Al-bawah, Ehab sharyan, Mohammed Al-Shehari, Khaled Badr, Saif Alaribi, Jafar A. Mohammed, Hadeel Alashwal.

**Investigation:** Mohamed Baklola, Naji Al-bawah, Ehab sharyan, Amjed Al-Jahmi, Saif Alaribi, Jafar A. Mohammed, Hadeel Alashwal, Mohammed Badi.

**Methodology:** Mohamed Baklola, Naji Al-bawah, Ehab sharyan, Mohammed Al-Shehari, Amjed Al-Jahmi, Hend O. Al-Sabri, Mohammed Badi.

**Project administration:** Mohamed Baklola, Naji Al-bawah, Ehab sharyan.

**Resources:** Hend O. Al-Sabri.

**Supervision:** Mohamed Baklola, Ehab sharyan.

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
