## [Decision Letter · Decision Letter 0]

12 Dec 2025

Dear Dr. sharyan,

plosone@plos.org. . . . A rebuttal letter that responds to each point raised by the academic editor and reviewer(s). You should upload this letter as a separate file labeled 'Response to Reviewers'.A marked-up copy of your manuscript that highlights changes made to the original version. You should upload this as a separate file labeled 'Revised Manuscript with Track Changes'.An unmarked version of your revised paper without tracked changes. You should upload this as a separate file labeled 'Manuscript'.

We look forward to receiving your revised manuscript.

Kind regards,

Mukhtiar Baig, Ph.D.

Academic Editor

PLOS One

Journal Requirements:

4. Please update your submission to use the PLOS LaTeX template. The template and more information on our requirements for LaTeX submissions can be found at http://journals.plos.org/plosone/s/latex.

5. Please provide a complete Data Availability Statement in the submission form, ensuring you include all necessary access information or a reason for why you are unable to make your data freely accessible. If your research concerns only data provided within your submission, please write "All data are in the manuscript and/or supporting information files" as your Data Availability Statement.

6. PLOS requires an ORCID iD for the corresponding author in Editorial Manager on papers submitted after December 6th, 2016. Please ensure that you have an ORCID iD and that it is validated in Editorial Manager. To do this, go to ‘Update my Information’ (in the upper left-hand corner of the main menu), and click on the Fetch/Validate link next to the ORCID field. This will take you to the ORCID site and allow you to create a new iD or authenticate a pre-existing iD in Editorial Manager.

7. Your ethics statement should only appear in the Methods section of your manuscript. If your ethics statement is written in any section besides the Methods, please move it to the Methods section and delete it from any other section. Please ensure that your ethics statement is included in your manuscript, as the ethics statement entered into the online submission form will not be published alongside your manuscript.

Reviewers' comments:

Reviewer's Responses to Questions

**Comments to the Author**

1. Is the manuscript technically sound, and do the data support the conclusions?

Reviewer #1: Yes

Reviewer #2: Partly

2. Has the statistical analysis been performed appropriately and rigorously?

Reviewer #1: Yes

Reviewer #2: Yes

3. Have the authors made all data underlying the findings in their manuscript fully available?

Reviewer #1: Yes

Reviewer #2: Yes

4. Is the manuscript presented in an intelligible fashion and written in standard English?

Reviewer #1: Yes

Reviewer #2: No

Reviewer #1: This manuscript represents a valuable and much-needed contribution to the fields of medical education and global health. It provides crucial baseline data from a challenging context. The methodological strengths are evident in the national scope and sample size. To fully realize its potential, the authors must address the key issues of the contradictory data availability statement, the misinterpretation of the low R-squared values, the typographical errors in tables, and the need for more strategic citations to frame the conflict context. With these revisions, this manuscript will be significantly strengthened and will make a strong candidate for publication.

Reviewer #2: This manuscript presents a timely and important national cross-sectional study examining the research involvement, knowledge, attitudes, and perceived barriers among undergraduate medical students in Yemen, a conflict-affected, low-resource setting. The topic is highly relevant to global health and medical education, particularly in contexts where local research capacity is critical for evidence-based practice. However, the manuscript would benefit from a Major Revisions to address several methodological and presentation issues before the manuscript can be considered for publication. you can find them below

Abstract

1) A total of 1,387 students completed a ………… this need to be in the results section.

2) I suggest removing the word ‘national,’ as the manuscript does not present information about the geographical distribution or representativeness of the ten included universities.

3) Furthermore, the data collection period (‘between November and December 2024’) should be omitted from the abstract to enhance clarity and brevity; this information is more suitable for the Methods section.

4) The reported 20.5% publication rate appears higher than expected for undergraduate medical students. It may be helpful to double-check the data or clarify the criteria used for defining ‘published research paper.

Introduction

1) guiding health policies, and driving innovation [1]……… Health policy” instead of “health policies” (more commonly used in academic contexts.

2) undergraduate medical students [4]. It also fosters a research-oriented ….. Combined your first two sentences more coherently by using “Early involvement in research also fosters a …”

3) Replaced “untapped” with “unrealized” for a slightly more formal academic tone.

4) ''Studies from the Arab region suggest that while students generally exhibit favorable attitudes toward research, their actual knowledge and participation remain low.'' Needs proper citations current references do not sufficiently support the claims.

5) “Remain low” TO “remain limited” (more academic tone)

6) “National level understanding” TO “comprehensive understanding”

7) “Addresses that need” TO “addresses this gap”

8) For clarity, the authors should clearly define the specific objectives of the study and, ideally, include a formal research question to guide the reader.

Methodology

This section contains the necessary components; however, it lacks clarity and logical organization. Several subsections appear fragmented or insufficiently integrated, making it difficult to follow the study procedures step by step. I recommend restructuring the section to present the study design, sampling approach, data collection procedures, study tool description, and statistical analysis in a more cohesive and sequential manner. Additionally, transitions between subsections should be improved to enhance readability and ensure that each methodological element is clearly linked to the overall study design.

1) Removed “nationwide” since representativeness across Yemen is not established. Change to "Multi-institutional".

2) “at ten universities in Yemen” TO “from ten universities in Yemen.”

3)"Using an estimated population of 10,000 medical students in Yemen.….." Based on what you claimed this estimation, this need to be referenced.

4) “Via the Raosoft calculator” TO “using the Raosoft calculator”

5) “yielding a target of” TO “resulting in a target sample of.”

6) "To account for clustering and design variability, …" The sampling section is confusing. The authors report using convenience sampling but simultaneously apply a design effect as if clusters were used. Convenience sampling does not require or justify a design effect. This should be corrected, and the sampling approach must be clearly defined and aligned with the sample size calculation.

7) "….yielding a target of 1,110 respondents. Ultimately, 1,387 students completed the survey…." (‘Ultimately, 1,387 students completed the survey’). This information belongs in the Results section, not in the Methods section. The Methods should describe only how the target sample size was determined, whereas the actual number of respondents should be reported in the Results. I recommend removing this sentence from the Methods and presenting it appropriately in the Results section.

8) The use of a non-probability convenience sampling method should be explicitly acknowledged as a limitation, as it affects generalizability.

9) The questionnaire is described in detail, but no information is provided regarding piloting or pre-testing in the Yemeni context. The authors should clarify whether the instrument was piloted to ensure cultural and contextual appropriateness.

10) The authors state that they assessed students’ research knowledge using ten multiple-choice questions; however, the specific items used to measure this domain are not provided in the manuscript or in the supplementary materials. Without seeing the actual knowledge questions, it is not possible to evaluate the validity, relevance, or appropriateness of the knowledge assessment. I recommend that the authors include the full set of knowledge items either within the Methods section, in an appendix, or as supplementary material to ensure transparency and allow readers to assess the rigor of the knowledge measurement.

Discussion

The Discussion section is generally good but could be strengthened by a deeper contextualization of the findings within the specific challenges of a conflict-affected country.

The discussion should more explicitly link the identified barriers (lack of training, insufficient faculty guidance, inadequate facilities) to the ongoing conflict and its impact on the higher education system in Yemen. For example, how has the conflict specifically affected faculty retention, access to international journals/databases, and the maintenance of research infrastructure?

.

Reviewer #1: No

Reviewer #2: No

---

## [Author Response · Author response to Decision Letter 1]

8 Jan 2026

Response to reviewers

We sincerely thank the editor and reviewers for their constructive and insightful comments, which have significantly improved the quality, clarity, and rigor of our manuscript. We have carefully addressed all comments and revised the manuscript accordingly. All changes have been incorporated into the revised version of the manuscript. Detailed point-by-point responses are provided below.

Reviewer #1

Comment 1:

This manuscript represents a valuable and much-needed contribution to the fields of medical education and global health. It provides crucial baseline data from a challenging context. The methodological strengths are evident in the national scope and sample size. To fully realize its potential, the authors must address the key issues of the contradictory data availability statement, the misinterpretation of the low R-squared values, the typographical errors in tables, and the need for more strategic citations to frame the conflict context.

Response:

We sincerely thank the reviewer for the positive assessment of the manuscript and for highlighting its relevance and methodological strengths. We have carefully addressed all the issues raised:

- Data availability: The data availability statement has been revised for clarity and consistency. The dataset is now publicly available through Figshare, and the following statement has been added to the manuscript:

“The datasets generated and/or analyzed during this study are publicly available in the Figshare repository at https://doi.org/10.6084/m9.figshare.30927329.”

- Interpretation of low R-squared values: We revised the Results and Discussion sections to explicitly acknowledge the low R² values and to clarify that, although certain predictors were statistically significant, they explain only a small proportion of the variance. This limitation is now clearly stated and discussed to avoid overinterpretation.

- Typographical errors in tables: All tables were carefully reviewed and corrected. This included resolving duplicated attitude items, correcting mislabeled variables, standardizing formatting, and improving table titles and footnotes.

- Conflict-context framing: We strengthened the Discussion by adding targeted citations and explicitly linking the identified research barriers to the impact of ongoing conflict on higher education, including faculty shortages, damaged infrastructure, limited funding, and restricted access to academic resources.

Reviewer #2

Comment 1 (In abstract):

A total of 1,387 students completed a ………… this need to be in the results section.

Response:

We thank the reviewer for this helpful suggestion. The abstract has been revised so that the number of participants is now reported exclusively in the Results section of the abstract, in line with standard reporting conventions.

Comment 2 (In abstract):

I suggest removing the word ‘national,’ as the manuscript does not present information about the geographical distribution or representativeness of the ten included universities.

Response:

We agree with this comment. The term “national” has been removed throughout the manuscript and replaced with “multi-institutional” where appropriate, including in the title, abstract, and main text.

Comment 3 (In abstract):

Furthermore, the data collection period (‘between November and December 2024’) should be omitted from the abstract to enhance clarity and brevity.

Response:

This suggestion has been implemented. The data collection period has been removed from the abstract and is now reported only in the Methods section.

Comment 4 (In abstract):

The reported 20.5% publication rate appears higher than expected for undergraduate medical students. It may be helpful to double-check the data or clarify the criteria used for defining ‘published research paper’.

Response:

We appreciate this observation. The data were rechecked and confirmed. To improve clarity, we have clarified in the Methods section that “publication” refers to authorship on a research paper published in a peer-reviewed journal, including supervised student publications. Additional contextual clarification has also been added in the Discussion.

In Introduction

Comment 5:

“Health policies” instead of “health policy”.

Response:

The wording has been corrected to “health policy.”

Comment 6:

Combine the first two sentences more coherently.

Response:

The sentences were merged and rephrased for improved coherence and flow, as suggested.

Comment 7:

Replace “untapped” with “unrealized”.

Response:

This wording change has been implemented.

Comment 8:

Claims about the Arab region require stronger citations.

Response:

We added additional, relevant citations from the Arab region to appropriately support this statement.

Comment 9:

Change “remain low” to “remain limited”.

Response:

The wording has been revised accordingly.

Comment 10:

Change “national level understanding” to “comprehensive understanding” and “addresses that need” to “addresses this gap”.

Response:

These wording changes have been incorporated, and the paragraph was revised to improve clarity and precision.

Comment 11:

Clearly define the study objectives and include a research question.

Response:

The Introduction now explicitly states the study objectives and includes a clear research question guiding the investigation.

In Methods

Comment 12:

Remove “nationwide” and use “multi-institutional”.

Response:

The term “nationwide” has been removed and replaced with “multi-institutional” throughout the Methods section.

Comments 13 and 14:

Clarify the population estimate and sample size calculation.

Design effect is not appropriate for convenience sampling.

Response:

We thank the reviewer for these important methodological comments. In response, the sample size determination section has been substantially revised to improve clarity and methodological rigor. Rather than relying on an uncertain national population estimate, the revised calculation is now based on the primary outcome of interest, namely the mean research knowledge score, using parameters derived from a previously published regional study among undergraduate medical students. This approach aligns with recommended practices for cross-sectional studies assessing continuous outcomes.

Additionally, we fully agree that the application of a design effect is not appropriate in the context of convenience sampling. Therefore, all references to a design effect have been removed, and the sampling approach and sample size calculation have been fully aligned and clarified. The revised Methods section now transparently reports a minimum required sample size of 370 participants, which was exceeded in the final sample.

Comment 15:

Actual number of respondents should be reported in Results, not Methods.

Response:

This has been corrected. The achieved sample size is now reported only in the Results section.

Comment 16:

Convenience sampling should be acknowledged as a limitation.

Response:

This limitation is now explicitly stated and discussed in the Limitations section.

Comment 17:

Clarify whether the questionnaire was piloted.

Response:

The Methods section has been updated to clarify that the questionnaire was reviewed and piloted among a small group of medical students to ensure clarity and contextual appropriateness.

Comment 18:

Provide the full set of knowledge assessment items.

Response:

We have addressed this important point by including the complete set of ten research knowledge multiple-choice questions as Supplementary material 1, with clearly defined correct answers and scoring methodology.

In Discussion

Comment 19:

Strengthen the discussion by explicitly linking barriers to the conflict-affected context.

Response:

The Discussion has been substantially revised to more explicitly connect identified barriers, such as lack of mentorship, limited infrastructure, and restricted access to academic resources, to the ongoing conflict and its impact on Yemen’s higher education system.

---

## [Decision Letter · Decision Letter 1]

15 Feb 2026

Dear Dr. sharyan,

Thank you for submitting your manuscript to PLOS ONE. After careful consideration, we feel that it has merit but does not fully meet PLOS ONE’s publication criteria as it currently stands. Therefore, we invite you to submit a revised version of the manuscript that addresses the points raised during the review process.

We look forward to receiving your revised manuscript.

Kind regards,

Mukhtiar Baig, Ph.D.

Academic Editor

PLOS One

**Journal Requirements:**

Reviewers' comments:

Reviewer's Responses to Questions

**Comments to the Author**

Reviewer #2: (No Response)

2. Is the manuscript technically sound, and do the data support the conclusions?

Reviewer #2: Partly

3. Has the statistical analysis been performed appropriately and rigorously?

Reviewer #2: Yes

4. Have the authors made all data underlying the findings in their manuscript fully available?

Reviewer #2: Yes

5. Is the manuscript presented in an intelligible fashion and written in standard English?

Reviewer #2: No

Reviewer #2: A) ABSTRACT

""A cross-sectional, multi-institutional survey was conducted among undergraduate medical students across ten universities in Yemen.""

….."""A validated online questionnaire was used to

collect data on demographic characteristics, prior research experience, attitudes toward research,

research knowledge, and perceived institutional barriers.""" this is further detailed and repeated statement which was previously stated in earlier in the introduction.

""Medical students in Yemen demonstrate generally positive attitudes toward

research;....."" this need to be revised since the study did not represent all ,

Over interpretation has been noticed in different sections in the manuscript this need to be revised.

Keywords: Please revise the keywords to make them more professional

B) Background

Authors may further sharpen the focus by more explicitly linking global disparities in research output to medical education and undergraduate research training, rather than research capacity in general.

Avoiding slightly rhetorical / generic phrases such as “not merely academic, strikingly uneven and perpetuates.....” and using more neutral language.

The transition between the first and second also third paragraphs could be smoother by briefly signaling the shift from regional evidence to the Yemeni context.

Ensure consistent use of terms such as research engagement, research participation, and research involvement throughout the background to avoid conceptual ambiguity.

The background would benefit from a clearer indication of how the findings could inform educational policy, curriculum development, or institutional strategies in Yemen.

C) Methods

Manuscripts should adhere to Observational studies criteria STROBE.

"""""This study employed a cross-sectional analytical design and was conducted among

undergraduate medical students from ten universities in Yemen.""""" What do you mean by "analytical"??

No details where stated regarding those "ten universities"

""....the survey was disseminated broadly, yielding a final sample of 1,387 respondents."" no need to be mentioned in this section. in this section you need to show how you selected sample population and Justification for Target Population.

""""Data collection was conducted using an online survey administered during the

study period."""" need for wording , further For methodological transparency, the authors should indicate the online platform through which the questionnaire was distributed.

The term “primary outcome of interest” should be clearly defined, and the specific outcome used for the sample size calculation should be explicitly stated.

The sample size calculation relies on data from a Jordanian population; the authors should explicitly state that the current study was conducted in Yemen and explain the rationale for using external estimates.

""Cronbach’s alpha values ranging from 0.70 to 0.80 across relevant sections......"" Authors need clarify for each section.

In the abstract authors have stated "".....and linear regression to identify factors associated with research knowledge and attitudes."" however in the methods section no information was indicated such a test has been preformed.

Did the authors preform normality test , this need to be stated conforming the normality of data distribution.

in the results section there were missing data in Table 3. in term of comparison of total knowledge and total attitude scores by question stating "Have you ever published your research paper in journals?"

in the discussion section authors need to void rhetorical / generic phrases and use more neutral language.

.

Reviewer #2: No

---

## [Author Response · Author response to Decision Letter 2]

11 Mar 2026

Response to Reviewer #2

We sincerely thank the reviewer for the constructive and insightful comments that helped improve the clarity and methodological rigor of our manuscript. We have carefully revised the manuscript in accordance with the suggestions provided. All changes have been incorporated into the revised manuscript.

A) Abstract

Comment 1

"A cross-sectional, multi-institutional survey was conducted among undergraduate medical students across ten universities in Yemen."

"A validated online questionnaire was used to collect data on demographic characteristics, prior research experience, attitudes toward research, research knowledge, and perceived institutional barriers."

This is a further detailed and repeated statement which was previously stated earlier in the introduction.

Response:

Thank you for this observation. The Methods section of the abstract has been revised to reduce redundancy and improve conciseness. The description of the questionnaire has been simplified while retaining the essential methodological details.

Comment 2

"Medical students in Yemen demonstrate generally positive attitudes toward research" this needs to be revised since the study did not represent all.

Response:

We agree with the reviewer and have revised this statement to avoid overgeneralization. The sentence has been modified to:

"Participating medical students demonstrated generally positive attitudes toward research."

Comment 3

Over interpretation has been noticed in different sections in the manuscript this needs to be revised.

Response:

Thank you for highlighting this point. We carefully reviewed the manuscript and revised several sentences throughout the abstract, discussion, and conclusion to ensure more cautious interpretation of the findings. Language suggesting causality or generalization has been replaced with more neutral phrasing (e.g., “suggest,” “may indicate,” and “among the surveyed students”).

Comment 4

Keywords: Please revise the keywords to make them more professional.

Response:

The keywords have been revised to improve clarity and relevance. The updated keywords are:

Undergraduate medical education; Research engagement; Medical students; Research barriers; Conflict-affected settings; Yemen.

B) Background

Comment 1

Authors may further sharpen the focus by more explicitly linking global disparities in research output to medical education and undergraduate research training, rather than research capacity in general.

Response:

We appreciate this suggestion. Additional sentences were added to the background section to explicitly connect global disparities in research output with the role of undergraduate medical education and research training in strengthening research capacity in low-resource settings.

Comment 2

Avoiding slightly rhetorical / generic phrases such as “not merely academic, strikingly uneven and perpetuates.....” and using more neutral language.

Response:

Thank you for this comment. The background section was carefully revised to remove rhetorical language and replace it with more neutral scientific wording. For example, the phrase referring to disparities being “not merely academic” was revised to a more neutral description of the potential implications for evidence-based decision-making.

Comment 3

The transition between the first and second also third paragraphs could be smoother by briefly signaling the shift from regional evidence to the Yemeni context.

Response:

We have revised the transitions between paragraphs to improve logical flow. Additional linking sentences were introduced to clearly move from the global perspective to regional evidence and finally to the specific context of Yemen.

Comment 4

Ensure consistent use of terms such as research engagement, research participation, and research involvement throughout the background to avoid conceptual ambiguity.

Response:

The manuscript has been revised to improve consistency in terminology. The term “research engagement” is now used consistently throughout the manuscript to describe students’ participation in research-related activities.

Comment 5

The background would benefit from a clearer indication of how the findings could inform educational policy, curriculum development, or institutional strategies in Yemen.

Response:

We agree with the reviewer and have added sentences in the background section emphasizing how the findings of this study may inform curriculum development, educational strategies, and institutional policies aimed at strengthening undergraduate research engagement in Yemen.

C) Methods

Comment 1

Manuscripts should adhere to Observational studies criteria STROBE.

Response:

The manuscript has been revised to better align with the STROBE guidelines for cross-sectional studies. Relevant methodological details have been clarified, and the STROBE checklist has been prepared for submission with the revised manuscript.

Comment 2

"This study employed a cross-sectional analytical design and was conducted among undergraduate medical students from ten universities in Yemen." What do you mean by "analytical"?

Response:

Thank you for this comment. To improve clarity, the term “analytical” has been removed. The sentence has been revised to state that the study employed a cross-sectional study design.

Comment 3

No details were stated regarding those "ten universities".

Response:

Additional information has been added in the Methods section indicating that the participating institutions included both public and private universities located in different regions of Yemen.

Comment 4

"...the survey was disseminated broadly, yielding a final sample of 1,387 respondents." No need to be mentioned in this section.

Response:

We agree with the reviewer. The statement referring to the final sample size has been removed from the sample size subsection and is now presented only in the Results section.

Comment 5

"Data collection was conducted using an online survey administered during the study period." Need for wording, further authors should indicate the online platform.

Response:

The Methods section has been revised to clarify the data collection procedure and specify the platform used. The revised text now indicates that the survey was administered electronically using Google Forms.

Comment 6

The term “primary outcome of interest” should be clearly defined.

Response:

We have clarified that the primary outcome used for the sample size calculation was the mean research knowledge score among undergraduate medical students.

Comment 7

The sample size calculation relies on data from a Jordanian population.

Response:

We have clarified this point in the Methods section and added a statement explaining that estimates from the Jordanian study were used because no prior large-scale studies assessing research knowledge among Yemeni medical students were available.

Comment 8

"Cronbach’s alpha values ranging from 0.70 to 0.80 across relevant sections." Authors need clarify for each section.

Response:

Thank you for this suggestion. The internal consistency values are now reported separately for each scale. The revised manuscript reports Cronbach’s alpha values of 0.78 for the attitude scale and 0.81 for the perceived barriers scale.

Comment 9

In the abstract authors have stated linear regression but no information in methods.

Response:

We appreciate this comment. The statistical analysis section has been revised to explicitly describe the use of multiple linear regression analyses to identify factors associated with research knowledge and attitude scores.

Comment 10

Did the authors perform normality test?

Response:

Yes. We have now added a statement in the statistical analysis section indicating that normality of continuous variables was assessed using the Shapiro–Wilk test and visual inspection of histograms prior to conducting parametric analyses.

Comment 11

In the results section there were missing data in Table 3 in term of comparison of total knowledge and total attitude scores by question stating "Have you ever published your research paper in journals?"

Response:

Thank you for identifying this omission. The variable “Have you ever published your research paper in journals?” has now been added to Table 3, and the corresponding statistical comparison has been included in the Results section.

Comment 12

In the discussion section authors need to avoid rhetorical / generic phrases and use more neutral language.

Response:

The Discussion section has been carefully revised to reduce rhetorical language and ensure a more neutral scientific tone. Several sentences were modified to avoid overinterpretation and to present the findings in a more cautious and evidence-based manner.

---

## [Editor Report · Decision Letter 2]

13 Apr 2026

Bridging the research gap in conflict-affected countries: a multi-institutional study on medical students' research involvement and barriers in Yemen

PONE-D-25-61392R2

Dear Dr. Sharyan,

We’re pleased to inform you that your manuscript has been judged scientifically suitable for publication and will be formally accepted for publication once it meets all outstanding technical requirements.

Kind regards,

Mukhtiar Baig, Ph.D.

Academic Editor

PLOS One
---

## [Editor Report · Acceptance letter]

PONE-D-25-61392R2

PLOS One

Dear Dr. sharyan,

I'm pleased to inform you that your manuscript has been deemed suitable for publication in PLOS One. Congratulations! Your manuscript is now being handed over to our production team.

Kind regards,

on behalf of

Professor Mukhtiar Baig

Academic Editor

PLOS One